# Genetic Diversity and Evaluation of Agro-Morphological Traits in Lettuce Core Collection

**DOI:** 10.3390/plants13243552

**Published:** 2024-12-19

**Authors:** Suyun Moon, Onsook Hur, Seong-Hoon Kim, Yoonjung Lee, Hyeonseok Oh, Jungyoon Yi, Ho-Cheol Ko, Hee-Jong Woo, Nayoung Ro, Young-Wang Na

**Affiliations:** 1National Agrobiodiversity Center, National Institute of Agricultural Sciences, Rural Development Administration, Jeonju 54874, Republic of Korea; sooym21@korea.kr (S.M.); shkim0819@korea.kr (S.-H.K.); yoon112@korea.kr (Y.L.); zzjiy@korea.kr (H.O.); naaeskr@korea.kr (J.Y.); hchko@korea.kr (H.-C.K.); woo001@korea.kr (H.-J.W.); 2Department of Crop Breeding, National Institute of Crop Science, Rural Development Administration, Wanju 55365, Republic of Korea; oshur09@korea.kr

**Keywords:** *Lactuca sativa*, germplasm, genetic diversity, population structure, agro-morphological characterization, genotyping by sequencing

## Abstract

Lettuce (*Lactuca sativa*) is a globally significant leafy vegetable, valued for both its economic and nutritional contributions. The efficient conservation and use of the lettuce germplasm are crucial for breeding and genetic improvement. This study examined the genetic diversity and population structure of a core collection of the lettuce germplasm using genotyping by sequencing (GBS). A total of 7136 high-quality single-nucleotide polymorphisms (SNPs) were identified across nine chromosomes. Population analysis through Bayesian clustering and discriminant analysis of principal components (DAPC) revealed three distinct genetic clusters. Cluster 2 exhibited the greatest genetic diversity (*He* = 0.29, *I* = 0.44), while Cluster 3 had high levels of inbreeding (*F* = 0.79). Agro-morphological trait evaluation further identified significant differences in leaf length, plant weight, and head height across clusters. These findings provide valuable insights into the genetic and phenotypic diversity of lettuce, facilitating the development of more robust breeding programs. Additionally, the core collection established in this study offers a representative subset of the lettuce germplasm for future genomic research and conservation efforts.

## 1. Introduction

The genus *Lactuca*, belonging to the family Asteraceae, comprises approximately 150 species that are primarily distributed across warm and temperate regions of the Northern Hemisphere, including Europe, Asia, Indonesia, and North and Central America [1]. Among these species, *Lactuca sativa* L., commonly referred to as lettuce, stands out due to its widespread cultivation and significant economic value [2]. As one of the most consumed leafy vegetables globally, lettuce reached a total production of approximately 27.1 million tonnes in 2022 [3]. Additionally, several wild *Lactuca* species, such as *L. serriola* L., *L. saligna* L., and *L. virosa* L., are of commercial interest, offering advantageous traits that are utilized as genetic resources in lettuce breeding programs [4,5]. Furthermore, *L. indica* L. has been traditionally used in folk medicine due to its medicinal properties [6,7].

Lettuce is classified into six horticultural types based on morphological characteristics: leaf, crisphead, romaine, butterhead, stem, and oilseed [8]. These morphological traits, especially those related to leaves and heads, are crucial for classification and directly impact the quality and yield of lettuce crops [9]. Environmental factors, including temperature, light quality, and nutrition, along with inherent genetic differences, significantly influence these morphological characteristics [8,10,11,12,13,14]. Consequently, understanding the genetic diversity within lettuce is essential for elucidating the interactions between genotypes and phenotypes, which is critical for improving various characteristics of the crop.

To assess genetic variation and diversity within *Lactuca* species, various molecular markers such as Restriction Fragment Length Polymorphism (RFLP), Random Amplified Polymorphic DNA (RAPD), Amplified Fragment Length Polymorphism (AFLP), and Simple Sequence Repeats (SSRs) have been employed [15,16,17,18,19]. However, these markers often provide limited genomic coverage, which can be insufficient for advanced breeding programs due to their limited number and uneven distribution across the genome [20]. Addressing these limitations, single-nucleotide polymorphisms (SNPs) have emerged as powerful tools in genomic analysis, offering extensive coverage and ubiquitous distribution across the genome [21,22]. The advent of high-throughput sequencing technologies has further facilitated the rapid and cost-effective discovery of SNPs [23]. Among these technologies, the genotyping-by-sequencing (GBS) approach has been widely applied in leafy crops, including spinach, chicory, cabbage, and kale [24,25,26,27,28]. This approach enables efficient genotyping of large populations, thereby advancing marker-assisted breeding and genomic studies in plant breeding programs [29,30].

In managing large germplasm collections, the presence of redundancy often poses challenges in terms of efficient management and evaluation [31]. The development of core collections has been proposed as a solution, offering a representative subset that captures the full range of genetic diversity within a germplasm collection [32,33]. These core collections facilitate in-depth genetic and phenotypic analyses while simplifying germplasm management and enhancing the efficiency of breeding programs [34].

Studies on developing core collections have been conducted in various plant species, particularly in economically important crops. For horticultural crops, core collections have been established in species such as spinach, amaranth, cabbage, and cauliflower [15,35,36,37]. However, research on the development of core collections for lettuce remains limited. One study aimed to enhance the understanding of genetic diversity assessment and core collection development by comparing various sampling methods using *L. sativa* accessions [38].

In this study, we investigated the genetic diversity and population structure within a 300-core collection of the lettuce germplasm. Additionally, we assessed the phenotypic diversity of this core collection by evaluating 17 qualitative and quantitative agro-morphological traits. Through this comprehensive analysis, we aim to enhance the understanding of genetic and phenotypic variability within the lettuce germplasm, thereby supporting future breeding efforts and improving resource management.

## 2. Materials and Methods

### 2.1. Plant Materials and DNA Extraction

A collection of the 2001 *Lactuca* germplasm, including 2 accessions of *L. altaica* Fisch. & C.A.Mey., 3 of *L. indica* L., and 1 accession each of *L. livida* Boiss. & Reut. and *L. saligna* L., along with 1982 accessions of *L. sativa* L. and 12 of *L. serriola* L., was utilized to develop the core collection. Detailed information on sample ID, species, origin, and accession type is provided in Appendix A. The lettuce was grown at the research farm of the National Agrobiodiversity Center (NAC), Rural Development Administration (RDA), Jeonju (35°49′18″ N, 127°08′56″ E), Republic of Korea, during 2022–2024, following the RDA-recommended lettuce cultivation method. Total genomic DNA was extracted from lyophilized seedling samples using a Genomic DNA Prep Kit (Inclone Biotech, Yongin, Korea) according to the manufacturer’s protocol.

### 2.2. GBS Library Preparation and Sequencing

The GBS libraries were constructed using the ApeKI restriction enzyme, following a previously established protocol with minor modifications [39]. The Illumina DNA Prep kit (Illumina Inc., San Diego, CA, USA) was utilized for library preparation according to the manufacturer’s instructions. The quality and fragment size distribution of the libraries were assessed using a Bioanalyzer equipped with a High Sensitivity DNA Chip (Agilent Technologies, Santa Clara, CA, USA). The GBS libraries were sequenced on the Illumina NovaSeq 6000 platform (Illumina Inc., San Diego, CA, USA).

### 2.3. SNP Discovery, Filtering, and Genotype Imputation

After sequencing, raw reads were demultiplexed based on barcode sequences using Stacks v2.60 [40] with the “process_radtags” function. The reads then underwent quality control with Trimmomatic v0.39 [41] to remove adapter sequences and low-quality reads, ensuring the retention of high-quality data for subsequent analysis. The cleaned reads were aligned to the reference genome of *Lactuca sativa* [42] using the Burrows-Wheeler Aligner (BWA) v0.7.17 [43]. Picard v2.26.4 (https://broadinstitute.github.io/picard/ accessed on 11 May 2022) was employed to add read group information, which is essential for proper duplicate marking and further processing.

To enhance the accuracy of SNP identification, the Genome Analysis Toolkit (GATK) v4.2.0.0 [44] was used to perform local realignment of reads. A summary of the sequencing results is presented in Appendix A. Following realignment, GATK HaplotypeCaller was utilized for variant calling, generating an initial set of SNPs and indels. The resulting variants were further filtered using VCFtools to obtain a high-confidence set of SNPs [45]. The filtering criteria included retaining only biallelic SNPs, requiring a minimum minor allele frequency (MAF) of 0.05, ensuring a mean read depth (meanDP) of at least 5, and excluding variants with more than 5% missing genotype data. The SNPs that were not assembled onto the chromosome were manually filtered out. The number of filtered SNPs and heterozygous genotype ratio for each chromosome are presented in Appendix A.

To further enhance the dataset’s completeness and utility for downstream analyses, genotype imputation was performed using the BEAGLE algorithm [46], with 1000 burn-in and 1000 main iterations.

### 2.4. Morphological Characterization

For the assessment of agro-morphological traits, ten plants per genotype were grown. Three representative plants, chosen based on their phenotypic characteristics, were evaluated for each genotype. Seventeen agro-morphological traits were recorded during field inspections, including plant growth type, leaf shape, leaf color, the density of leaf margin incisions, leaf blistering, head formation, head shape in the longitudinal section, head size, head density, leaf length (cm), leaf width (cm), leaf thickness (mm), plant weight (g), head height (cm), head diameter (cm), days to 50% bolting, and days to 50% flowering (Appendix A). With the exception of days to 50% bolting and flowering, all traits were investigated at harvest stage. All plants were evaluated at the same time to ensure consistency. Qualitative traits related to the head, such as head shape in the longitudinal section, head size, and head density, were measured only in cases where the head formation trait exhibited semi-heading or heading types. To ensure accuracy and consistency in the data, accessions that exhibited poor growth were excluded from quantitative trait measurements. For each trait, the values were reported as the mean of triplicate determinations.

### 2.5. Development of Lettuce Core Collection

To develop the core collection of 2001 *Lactuca* germplasm, we analyzed 7136 high-quality SNPs obtained through GBS and stringent filtering, along with 9 qualitative morphological traits. We focused on these nine traits because quantitative traits are often heavily influenced by environmental factors, which can obscure the genetic relationships among accessions. Initially, 400 germplasm accessions were selected based on the genetic distance matrix using the R package Core Hunter [47]. From these 400 lettuce germplasm accessions, a representative subset of 300 accessions was selected as the core collection by considering qualitative traits and applying the advanced maximization (M) strategy, which was implemented through a modified heuristic algorithm in PowerCore software [48]. To compare the genetic diversity between entire and core collections based on qualitative agro-morphological traits, the Shannon–Weaver diversity index (*I’*), Nei’s genetic diversity index (*H’*), and genetic evenness (*J*) were calculated, following Hennink and Zeven [49] and Pielou [50].

### 2.6. Analysis of Population Structure

Bayesian-based clustering was performed using STRUCTURE v2.3.4 [51] by testing ten independent runs with K ranging from 1 to 10. Each run included a burn-in period of 10,000 iterations followed by 10,000 Markov Chain Monte Carlo (MCMC) iterations, assuming the admixture model. The optimal number of subpopulations and subtle clustering patterns were identified using StructureSelector [52] via the ∆K method [53]. A membership coefficient (q) greater than 0.6 was used to assign samples to specific clusters, while samples with membership coefficients < 0.6 were classified as ‘genetically admixed’. The admixture proportions of each lettuce accession, estimated by STRUCTURE, were visualized using STRUCTURE PLOT v2.0 [54].

The genetic structure was further analyzed by the Discriminant Analysis of Principal Components (DAPC) using the R package adegenet [55]. The “find.clusters” function was used to detect the number of genetic clusters in the population. The best number of subpopulations in the DAPC was indicated by an elbow in the curve of the Bayesian information criterion (BIC). A cross-validation function “xvalDapc” was used to confirm the correct number of PCs to be retained and then the analysis was rerun with this number.

Genetic distances between pairs of accessions were calculated, and an unrooted unweighted pair group method with arithmetic means (UPGMA) was constructed using TASSEL5 [56]. The resulting unrooted UPGMA tree was visualized using Interactive Tree Of Life (iTOL) v6 [57].

### 2.7. Analysis of Molecular Variance (AMOVA) and Genetic Diversity Indices

The number of subpopulations determined based on the DAPC analysis was used for calculating AMOVA, the PhiPT index, and Nei’s genetic distance using GenAlEx v6.51b2 [58]. To further verify the genetic differentiation, the number of different (*Na*), and effective alleles (*Ne*), the expected heterozygosity (*He*), the fixation index (*F*), Shannon’s information index (*I*), and pairwise PhiPT values were calculated for each population. To determine whether variance component partitioning was significant, we estimated the probability values by using 999 permutations.

### 2.8. Statistical Analysis

Data summarization and descriptive statistics on agro-morphological data were performed using the R v4.4.1 [59]. Significant differences between mean values were computed using the Student–Newman–Keuls multiple comparison test (α = 0.05).

## 3. Results

### 3.1. Distribution of SNPs in Lactuca sativa Genome

Approximately 6.9 billion reads were generated from the 2001 *Lactuca* germplasm accessions using the Illumina NovaSeq 6000 platform, with an average mapping depth of 7.52× per accession. A summary of these sequencing results is provided in Appendix A. For the identification of polymorphic loci, only SNPs with a minor allele frequency (MAF) of ≥0.05 and a mean depth of coverage (meanDP) of ≥5 were considered. A total of 7136 high-quality SNPs were physically mapped across nine chromosomes (Appendix A). Chromosome 4 harbored the highest number of SNPs (1076 SNPs), whereas chromosome 6 contained the fewest (451 SNPs). These high-quality SNPs were distributed across the *L. sativa* genome, with variable densities observed along different chromosomal regions, as illustrated in Figure 1.

### 3.2. Development of Core Collection

Genetic distances between 2001 *Lactuca* accessions were calculated using Core Hunter based on the 7136 SNPs, leading to the initial selection of 400 representative accessions. Among these 400 accessions, accessions that were admixed or exhibited poor growth were excluded. Subsequently, qualitative phenotypic data were incorporated to refine this selection using the M strategy, resulting in a final core collection of 300 accessions. PCA of these 300 accessions in the core collection showed nearly identical patterns to those of the 2001 accessions in the entire collection (Figure 2A). The resulting core collection was then used for further analyses.

Phenotypic characterization of 17 agro-morphological traits was performed for the lettuce core collection. The frequency and relative percentage of qualitative agro-morphological traits are summarized in Table 1. The majority of the accessions were categorized as leaf type, with lesser representation from other types such as romaine, butterhead, crisphead, and stem. The predominant leaf shapes within the germplasm were elliptic and broad elliptic. Regarding leaf color, green and red were the two most common colors observed. Most accessions displayed medium blistering on leaves, followed by weak and strong blistering. In terms of head formation, accessions were classified as non-heading, heading, or semi-heading types. Traits such as head shape in the longitudinal section, head size, and head density were evaluated for the semi-heading and heading types. Circular head shapes were the most common, followed by elliptic and broad elliptic. Medium-sized heads were most prevalent, and head density ranged from absent and loose to medium, dense, and very dense categories. Additionally, the genetic diversity and evenness between the entire and core collections based on qualitative agro-morphological traits were compared (Appendix A). The mean values for the Shannon–Weaver diversity index (*I’*) and Nei’s genetic diversity index (*H’*) in the core collection (0.94 and 0.70, respectively) were similar to those of the entire collection (1.09 and 0.78, respectively). Moreover, the mean genetic evenness (J) values for the core and entire collections were also comparable (0.61 and 0.69, respectively). These results demonstrate that the developed core collection adequately represents the morphological characteristics of the entire collection.

For quantitative agro-morphological data, the minimum, maximum, average, standard deviation, and coefficient of variation (CV) values are presented in Table 2. The average values for leaf length, leaf width, and leaf thickness were 25.32 cm, 18.65 cm, and 0.21 mm, respectively. Plant weight showed the widest variation, ranging from 118.00 g to 1940.33 g. Regarding the height and diameter of the head, the average values were 28.92 cm and 18.87 cm, respectively. The average number of days to bolting and flowering were 79.20 and 102.58 days, respectively. Five of the eight traits had relatively high CV values above 20%, with the highest CV observed for plant weight (38.36%) and the lowest for days to flowering (8.75%).

### 3.3. Population Structure and Genetic Relationships

To investigate the population structure of the lettuce core collection, the Bayesian-based clustering analysis was conducted. The ΔK analysis showed that the highest ΔK value was K = 3, suggesting the presence of three main populations in the lettuce core collection (Figure 2B,C). Population 1, 2, and 3 consisted of 72, 130, and 24 accessions, respectively. Seventy-four accessions with a membership coefficient below 0.6 were identified as a genetically admixed population. Furthermore, population structure analysis confirmed that the three clusters observed in the core collection are also present in the whole collection (Appendix A).

The lowest BIC value was found at K = 4, but elbowed at K = 3, in the DAPC (Figure 3A). The cluster analysis classified the accessions into three clusters following the elbow method. The result of cross-validation for the remaining PCs was optimized to the 20 first components, where it had highest mean successful assignment and the lowest root mean squared error (RMSE) (Figure 3B). For the subsequent DAPC, twenty PCs and two discriminant functions were retained. The resulting scatterplot of the lettuce core collection is shown in Figure 3C. The grouping assignment for individual accessions by the DAPC is listed in Appendix A. Cluster 1, 2, and 3 consisted of 91, 170, and 39 accessions, respectively. Among the three clusters, cluster 2 had the highest number of admixed individuals (39 accessions). The unweighted pair group method with arithmetic means (UPGMA) analysis of the lettuce core collection is presented in Figure 4. The resulting phylogenetic tree reveals significant genetic diversity both within and between the clusters. For further genetic analysis, the clusters of the DAPC were used because it separated the lettuce core collections from the populations of the Bayesian-based clustering in more detail.

### 3.4. Genetic Differentiation and Genetic Diversity Across Populations

The genetic variability within and among populations was assessed by using analysis of molecular variance (AMOVA). The AMOVA results indicated that 14% of the total genetic variation was partitioned among populations, while a substantial 86% of the variation was found within populations (Table 3). The genetic differentiation among populations was found to be moderate (with PhiPT = 0.14), with significance at *p* < 0.001 and a gene flow value (Nm) of 1.56. This result suggests that genetic differentiation is more pronounced within subpopulations than among them. The pairwise PhiPT genetic distance values ranged from 0.04 between Cluster 1 and Cluster 2 to 0.12 between Cluster 1 and Cluster 3, indicating varying degrees of genetic divergence between the clusters (Table 4). Notably, the PhiPT values were statistically significant (*p* < 0.001) for all pairwise comparisons between subpopulations, underscoring the genetic distinctiveness of each cluster.

The grand mean values for the number of different alleles (*Na*) and the number of effective alleles (*Ne*) across the three subpopulations were 1.96 and 1.45, respectively (Table 5). The mean values for the entire population in terms of the expected heterozygosity (*He*), the unbiased diversity index (*uHe*), and Shannon’s information index (*I*) were 0.28, 0.28, and 0.43, respectively. Among the three clusters, Cluster 2 exhibited the highest genetic diversity, as indicated by its slightly higher values for the *He* (0.29), *uHe* (0.29), and *I* (0.44), compared to Clusters 1 and 3. The fixation index (*F*), which measures the degree of inbreeding within each population, ranged from 0.77 to 0.79, indicating a consistently high level of genetic structuring across the clusters.

### 3.5. Phenotypic Variation Analysis Across Population

The phenotypic variation among the three clusters identified in the lettuce core collection was assessed using various agro-morphological traits, as shown in Figure 5 and Appendix A. Significant differences were observed across the clusters for several traits, indicating distinct phenotypic profiles. Cluster 3 exhibited the longest average leaf length (29.80 cm), which was significantly greater than that of Clusters 1 (21.92 cm) and 2 (26.54 cm) (*p* < 0.001). However, leaf width and thickness did not show significant differences across the clusters. Cluster 2 had a higher average plant weight compared to the other two clusters, with a value of 571.96 g. Significant variation was also observed in head height, with Cluster 3 having the tallest heads on average (34.26 cm), while head diameter did not differ significantly across the clusters. Similarly, the average number of days to bolting and flowering were similar across all clusters.

## 4. Discussion

Studying genetic diversity and population genetics is essential for the conservation and sustainable use of crop germplasms [60,61]. In this study, a core collection of the *Lactuca* germplasm was developed by integrating high-density genotypic data with agro-morphological data. This approach enhances research efficiency and supports breeding programs, ensuring that the genetic diversity required for future agricultural innovation is preserved and utilized effectively [62,63]. Our findings provide valuable insights for future breeding efforts, laying the foundation for genome-wide association studies (GWASs) and marker-assisted selection in lettuce.

GBS analysis identified 7136 high-quality SNPs distributed across nine chromosomes, offering a comprehensive overview of genetic variation. These SNPs were crucial in revealing the genetic relationships among *Lactuca* accessions. The variability in SNP counts across studies reflects methodological differences, such as filtering stringency and data handling. Missing data are a common challenge in GBS-based research, often resulting in a significant reduction in the number of SNPs [64,65]. For instance, Park et al. [66] and Lee et al. [67] reported vastly different SNP numbers due to varied filtering criteria, particularly in the handling of missing genotype data. By applying stringent criteria, we ensured data reliability while minimizing the potential loss of data.

The first core collection in plants was established in chickpea based on phenotypic data and geographical distribution [68]. The evolution of core collections from simple phenotypic assessments to advanced genetic integrations exemplifies progress in germplasm management [69,70,71,72,73,74]. Given that using only genotypic or phenotypic information for establishing core collections may not efficiently capture the full genetic diversity of a species [75], we constructed a lettuce core collection through the integration of SNP data and agro-morphological data. Additionally, our lettuce core collection was refined by excluding admixed and poorly performing accessions, because admixed accessions may introduce genetic redundancy, while those with poor growth performance can distort the evaluation metrics of the core collection, potentially compromising breeding outcomes [34,76]. This refinement not only preserves genetic variation but also enhances its utility for breeding [77,78].

Although all wild *Lactuca* species accessions were excluded from the final core collection due to genetic redundancy and poor growth performance, additional accessions from other *Lactuca* species were initially included to capture a broader spectrum of genetic diversity. Based on their cross-compatibility and the fertility of F1 hybrids, *Lactuca* species can be categorized into the lettuce gene pool [79]. The primary gene pool, which is fully interfertile, includes *L. sativa*, *L. altaica*, and *L. serriola*, while the secondary gene pool consists solely of *L. saligna* [5]. Historically, *L. altaica* has been considered conspecific with *L. serriola* [17,80]. However, recent findings on morphological traits, phytochemical profiles, and plastid phylogeny suggest distinct differences, warranting further phylogenetic analyses to clarify its taxonomic status [81,82]. Similarly, *L. livida* is regarded as conspecific with *L. virosa* due to the continuity of their morphometric and biological traits [83]. *L. virosa*, in turn, is classified as a member of the tertiary gene pool. For *L. indica*, it has been reported as relatively distantly related to *L. sativa*, despite some shared genetic characteristics [15,81]. However, the limited sample sizes of wild species compared to *L. sativa* may introduce bias in these conclusions. To achieve a comprehensive understanding of genetic diversity and evolutionary relationships, broader sampling of wild *Lactuca* species is necessary.

Population structure analysis, employing Bayesian clustering and discriminant analysis of principal components (DAPC), identified three distinct genetic clusters (Figure 2 and Figure 3). Similar results have been reported with target region amplification polymorphism markers that were used to analyze fifty-four cultivars [84]. These clusters exhibited varying levels of genetic diversity, with Cluster 2 displaying the highest genetic heterogeneity and Cluster 3 emerging as the most genetically distinct. Although the UPGMA analysis also assigned the lettuce core collection to three major clusters, discrepancies were observed when compared with Bayesian clustering and DAPC regarding the assignment of accessions to specific groups (Figure 4). The distance-based nature of the UPGMA method may not effectively address compositional heterogeneity and rate variation, potentially leading to systematic clustering errors. In contrast, model-based approaches such as Bayesian clustering and DAPC compensate for these complexities, ensuring more reliable results [85,86]. The presence of admixed accessions, especially among cultivated varieties, suggests extensive hybridization and gene flow, likely driven by breeding practices and natural cross-pollination events within *L. sativa* and occasionally through hybridization with wild relatives such as *L. serriola*, *L. saligna*, and *L. virosa* [87,88].

The AMOVA revealed that the majority of genetic diversity was found within populations rather than between them (Table 3). This pattern of higher within-population variance compared to among-population variation has also been observed in several other crops, such as rutabaga, rice, and wheat [89,90,91]. Wright [92] noted that gene flow (Nm) values greater than 1 generally indicate sufficient migration to prevent significant genetic divergence between populations. Thus, the high gene flow (Nm = 1.56) and relatively low genetic differentiation (PhiPT = 0.14) observed in this study likely explain the distribution of genetic diversity within and among populations. Cluster-specific genetic distances further emphasize the influence of breeding history and local adaptation, with Cluster 3’s distinctiveness likely stemming from its high proportion of landraces, which have adapted to specific environments.

Significant phenotypic variations in traits such as leaf length, plant weight, and head height were observed, highlighting the potential for targeted breeding. In contrast, traits like leaf width, head diameter, and the average number of days to bolting and flowering exhibited limited variation between clusters, suggesting that these traits may be less influenced by the genetic diversity observed (Figure 5, Appendix A). The observed phenotypic diversity provides a foundation for future studies, including QTL mapping and GWASs, to identify loci linked to agronomically important traits.

Previous research has already demonstrated the value of these approaches in uncovering genetic determinants of traits, including post-harvest shelf life, disease resistance, and bolting time [67,93,94,95,96,97,98]. Incorporating such methods would provide a more refined understanding of the genetic mechanisms underlying important traits, potentially opening new avenues for targeted breeding strategies.

## 5. Conclusions

This study demonstrates that integrating genotypic data with phenotypic data can be effectively used to develop a core collection of the *Lactuca* germplasm and to identify significant genetic diversity and population structure within it. The identification of three genetically distinct clusters, along with significant phenotypic variations, particularly in leaf length, plant weight, and head height, provides valuable insights for lettuce breeding programs. The moderate genetic differentiation among clusters and high within-population diversity suggest strong gene flow between clusters, further supporting the utility of this core collection for future genome-wide association studies and marker-assisted selection. These findings enhance our understanding of lettuce genetic diversity and will contribute to improved cultivar development and the conservation of genetic resources.

## Figures and Tables

**Figure 1 plants-13-03552-f001:**
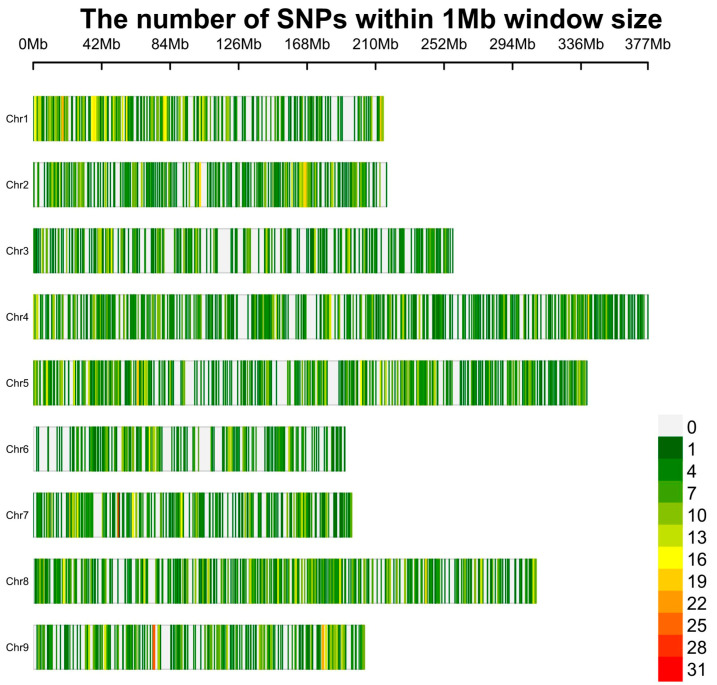
The distribution of 7136 high-quality SNPs across the 9 *Lactuca sativa* chromosomes. The horizontal axis represents the chromosome length in Mb. Different colors correspond to SNP density.

**Figure 2 plants-13-03552-f002:**
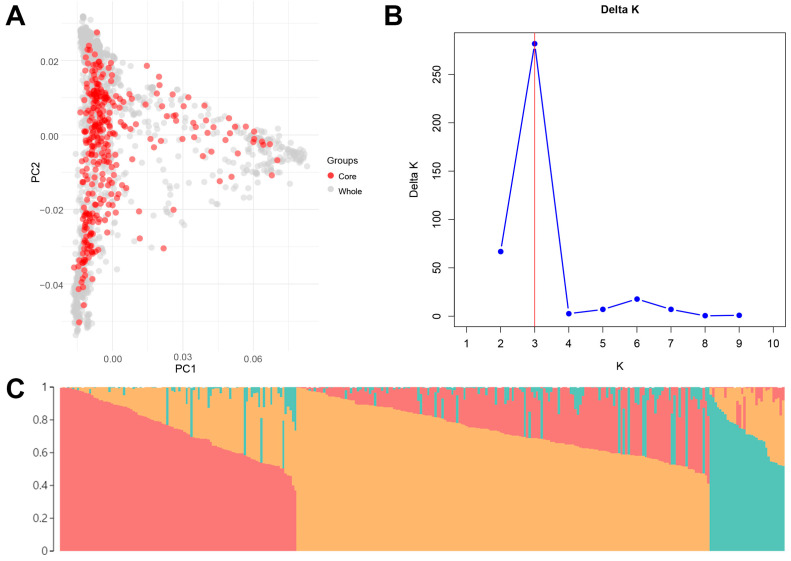
The development and population structure analysis of the lettuce core collection. (**A**) Principal component analysis (PCA) of the core collection. The red dots represent accessions in the core collection, while the gray dots represent accessions not included in the core collection. (**B**) Delta K values for different assumed population numbers (K) in the STRUCTURE analysis. (**C**) The inferred population structure for K = 3. Individual membership coefficients were sorted within each cluster.

**Figure 3 plants-13-03552-f003:**
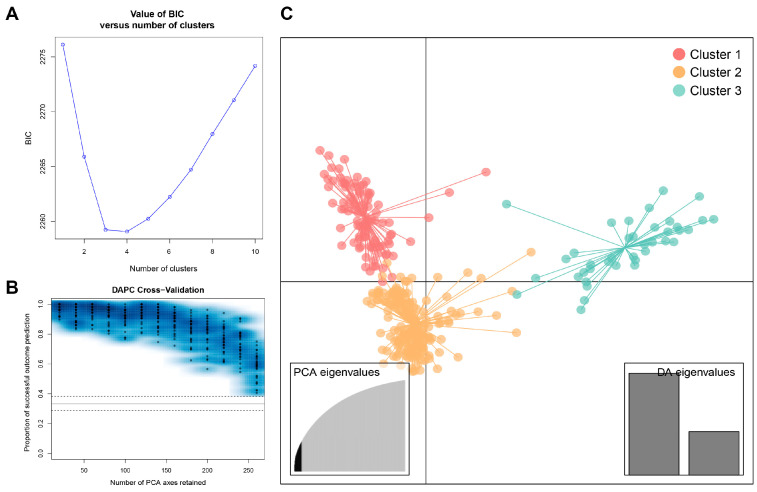
Discriminant analysis of principal components (DAPC) results of lettuce core collection. (**A**) Inference of optimal number of clusters for lettuce core collection based on BIC values. (**B**) DAPC cross-validation determining optimal number of principal components (PCs) retained for analysis of three predefined groups. (**C**) Scatterplot of individuals projected onto first twenty principal components (PCs) from DAPC. The colors represent the three clusters identified: red (Cluster 1), orange (Cluster 2), and teal (Cluster 3). Eigenvalues from analysis are displayed in inset.

**Figure 4 plants-13-03552-f004:**
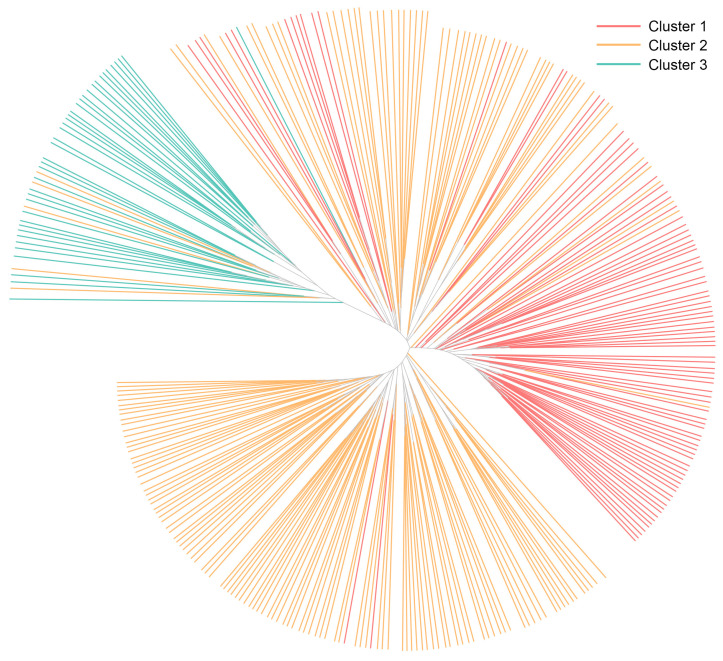
The unrooted UPGMA phylogenetic tree based on the genetic distance matrix representing the grouping of the lettuce core collection.

**Figure 5 plants-13-03552-f005:**
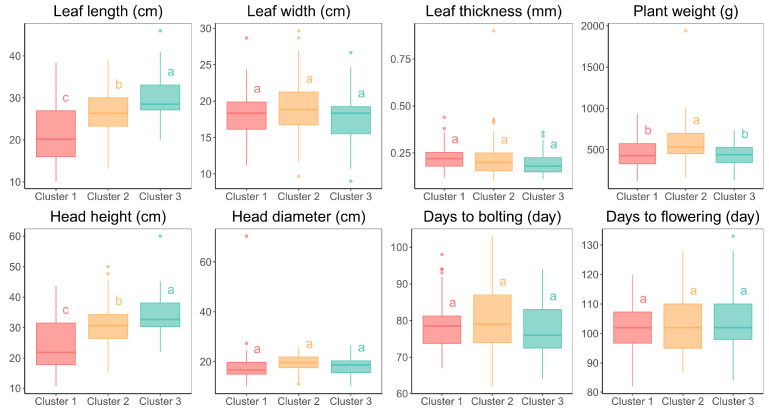
Boxplots showing the variation in quantitative agro-morphological traits across clusters. Different letters indicate significant differences between groups (*p* < 0.05).

**Table 1 plants-13-03552-t001:** Summary of qualitative agro-morphological traits of lettuce core collection.

Traits	Frequency	Percentage	Traits	Frequency	Percentage
Plant growth type			Leaf blistering		
Leaf	186	62.00	Absent	7	2.33
Romaine	27	9.00	Weak	90	30.00
Butterhead	47	15.67	Medium	152	50.67
Crisphead	34	11.33	Strong	50	16.67
Stem	6	2.00	Very strong	1	0.33
Leaf shape			Head formation		
Triangular	8	2.67	Non-heading	190	67.38
Narrow elliptic	10	3.33	Semi-heading	6	2.13
Elliptic	106	35.33	Heading	86	30.50
Broad elliptic	89	29.67	Head shape in		
Circular	39	13.00	longitudinal section		
Obovate	29	9.67	Elliptic	22	25.00
Oblate	7	2.33	Broad elliptic	20	22.73
Broad oblate	5	1.67	Circular	40	45.45
Broad obtrullate	7	2.33	Oblate	5	5.68
			Other	1	1.14
Leaf color			Head size		
Yellow	3	1.00	Very small	9	10.11
Green	196	65.33	Small	31	34.83
Greyish green	4	1.33	Medium	34	38.20
Cyan	13	4.33	Large	15	16.85
Red	84	28.00	Head density		
Density of			Absent	18	20.22
leaf margin incisions			Loose	12	13.48
Absent or sparse	219	73.00	Medium	17	19.10
Medium	66	22.00	Dense	16	17.98
Dense	15	5.00	Very dense	26	29.21

**Table 2 plants-13-03552-t002:** Summary of quantitative agro-morphological traits of lettuce core collection.

Traits	No. of Germplasm	Min	Max	Average	SD	CV (%)
Leaf length (cm)	262	10.17	46.00	25.32	6.46	25.51
Leaf width (cm)	262	9.00	29.67	18.65	3.40	18.23
Leaf thickness (mm)	262	0.10	0.90	0.21	0.08	38.10
Plant weight (g)	262	118.00	1940.33	518.60	198.96	38.36
Head height (cm)	261	10.67	60.00	28.92	7.65	26.45
Head diameter (cm)	262	10.00	70.33	18.87	4.78	25.33
Days to bolting	262	62.00	103.00	79.20	8.11	10.24
Days to flowering	262	82.00	133.00	102.58	8.98	8.75

CV, coefficient of variation.

**Table 3 plants-13-03552-t003:** Analysis of molecular variance (AMOVA), genetic differentiation (PhiPT), and gene flow.

Source	df	SS	MS	Est. Var.	%	PhiPT	Nm
Among Populations	2	104,454.03	52,227.02	569.25	14%	0.14 **	1.56
Within Populations	297	1,056,730.75	3558.02	3558.02	86%		
Total	299	1,161,184.78		4127.27	100%		

** *p*-value < 0.001. df, degree of freedom; SS, sum of squares; MS, mean of squares; Est. Var., estimated variance; %, percentage of genetic variation.

**Table 4 plants-13-03552-t004:** Pairwise PhiPT and genetic distances between populations.

Subpopulation	Cluster 1	Cluster 2	Cluster 3
Cluster 1	-	0.04	0.12
Cluster 2	0.09	-	0.09
Cluster 3	0.25	0.18	-

PhiPT values (above diagonal) and Nei’s genetic distance (blow diagonal) between.

**Table 5 plants-13-03552-t005:** Genetic diversity parameters of lettuce core collections.

Population	*N*	*Na*	*Ne*	*He*	*uHe*	*I*	*F*
Cluster 1	91	1.98	1.43	0.26	0.27	0.41	0.77
Cluster 2	170	1.99	1.47	0.29	0.29	0.44	0.79
Cluster 3	39	1.90	1.47	0.28	0.28	0.42	0.79
Mean		1.96	1.45	0.28	0.28	0.43	0.78

*N*, sample size; *Na*, mean number of different alleles; *Ne*, mean number of effective alleles; *He*, expected heterozygosity; *uHe*, unbiased diversity index; *I*, Shannon’s information index; *F*, fixation index.

## Data Availability

The SNP data presented in this study are deposited in the National Agricultural Biotechnology Information Center repository under accession number NV-0906.

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
