# Peer review of "Genetic Diversity and Evaluation of Agro-Morphological Traits in Lettuce Core Collection"

_plants, 2024, doi:10.3390/plants13243552_

Round 1
Reviewer 1 Report
Comments and Suggestions for Authors
This manuscripts report and original an interesting work on the construction and evaluation of a core collection on lettuce. In my opinion it deserves publication upon some modifications. In particular, authors need to compare the variability of the core collection with the whole collection of cultivars.
Introduction
Some more information on previous core collections on lettuce (if any) and in other similar plant species are needed.
Materials and methods
In line 329 it is stated that “For each trait, the values were reported as the mean of triplicate determinations.”. But authors need to indicate how many plants per genotype were evaluated, at what plant age the traits were evaluated and if all the plants were evaluated at the same time or not.
You also have to explain why you have measured 17 agro-morphological traits, but you only have included 9 qualitative traits for developing the core collection.
You have to mention the origin of the 7136 SNPs that you have used.
Results
Paragraph of lines 108 to 124. You do not have to include the figures of the percentages as the already appear in Table 1.
In my opinion it is compulsory that you compare the percentages of the full lettuce collection with the ones of the core collection.
The same for the Figures 2 and 3C. You should compare the distribution of the variability of the whole collection versus the core collection. Are the three clusters observed in the core collection also found in the whole collection?
In my opinion, Figure 4 does not add any further result respect to Figure 3. Therefore, Figure 4 could be removed without any loss of meaning.
Reviewer 2 Report
Comments and Suggestions for Authors
This study seems interesting and deserves to be considered. However, some points need further clarification.
I cannot see further discussion about the other species (different from Lactuca sativa) mentioned in the Introduction, M&M and Results.
The study focused on L. sativa, but what is the purpose of the additional accessions to the other mentioned species?
Moreover, L. altaica is considered a synonym of L. serriola, while L. livida is a synonym of L. virosa. This information should be provided in the article and properly discussed, with appropriate bibliographical references. The purpose of the accessions to the other species should be addressed, clearly presented and discussed. In the Table S1 all the records are reported as L. sativa, without mentioning the other species.
In presenting the species used for this study, (exactly) the same information is repeated, compare lines 78-80 and 281-283. The position of the Material & Methods sections could be moved before Results in order to respect the standard IMRaD format. This would allow to avoid repeating the same information at the beginning of Results.
In this case, please add the complete authorship to the names mentioned in the M&M section.
... of L. altaica , 3 of L. indica, and 1 accession each of L. livida and L. saligna, along 79 with 1,982 accessions of L. sativa and 12 of L. serriola (Table S1) à … of L. altaica Fisch. & C.A.Mey., 3 of L. indica L., and 1 accession each of L. livida Boiss. & Reut. and L. saligna L., along 79 with 1,982 accessions of L. sativa L. and 12 of L. serriola L. (Table S1)
Furthermore, it would be good practice to add the authorship to the species the first time they are cited in the article, to frame them unambiguously
Line 34 Lactuca sativa à Lactuca sativa L.
Line 37 L. serriola, L. saligna, and L. virosa à L. serriola L., L. saligna L., and L. virosa L.
Line 39 L. indica à L. indica L.
Round 2
Reviewer 2 Report
Comments and Suggestions for Authors
Authors evaluated the comments and made the requested changes.
I have no further comments and in my opinion the manuscript can now be considered for publication.